# Sea-Crossing Bridge Detection in Polarimetric SAR Images Based on Windowed Level Set Segmentation and Polarization Parameter Discrimination

**Chun Liu** [1] , **Chao Li** [2] **, Jian Yang** [3,*] **and Liping Hu** [2]

1    School of Software, Northwestern Polytechnical University, Xi'an 710072, China
2    Science and Technology on Electromagnetic Scattering Laboratory, Beijing 100854, China
3    Department of Electronic Engineering, Tsinghua University, Beijing 100084, China
*    Correspondence: yangjian_ee@tsinghua.edu.cn

**Abstract:** As sea-crossing bridges are important hubs connecting separated land areas, their detection in SAR images is of great significance. However, under complex scenarios, the sea surface conditions, the distribution of coastal terrain morphologies, and the scattering components of different structures in the bridge area are very complex and diverse, which makes the accurate and robust detection of sea-crossing bridges difficult, including the sea–land segmentation and bridge feature extraction on which the detection depends. In this paper, we propose a polarimetric SAR image detection method for sea-crossing bridges based on windowed level set segmentation and polarization parameter discrimination. Firstly, the sea and land are segmented by a proposed windowed level set segmentation method, which replaces the construction of the level set segmentation energy function based on the isolated pixel distribution with a joint distribution of pixels in a certain window region. Secondly, water regions of interest are extracted by a proposed water region merging algorithm combining the distances of the water contour and polarization similarity parameter. Finally, the bridge regions of interest (ROIs) are extracted by merging close water contours, and the ROIs are discriminated by the polarimetric parameters of the polarization entropy and scattering angle. Experimental results using multiple AirSAR, RADARSAT-2, and TerraSAR-X quad-polarization SAR data from the coastal areas of San Francisco in the USA, Singapore, and Fuzhou, Fujian, and Zhanjiang, Guangdong, in China show that the proposed method can achieve 100% detection of sea-crossing bridges in different bands for different scenes, and the accuracy of the intersection of the ground-truth (IoG) index of bridge body recognition can reach more than 85%. The proposed method can improve the detection rate and reduce the false alarm rate compared with the traditional spatial-based method.

**Keywords:** bridge detection; polarimetric synthetic aperture radar (PolSAR); sea-crossing bridge; level set segmentation; water merging; polarimetric parameter extraction

## 1. Introduction

Bridges are some of the most important stationary artificial buildings. The automatic detection of bridge targets in SAR images is important for disaster prevention, automatic navigation, and terrain mapping. Since the sea-crossing bridges are transportation hubs connecting different land areas, their detection is even more significant. For example, when natural disasters such as floods and earthquakes occur, the bridge recognition based on the SAR image can be used to assess the disaster situation and formulate disaster relief plans. When a flying platform deviates from its orbit during the flight, attitude correction can be achieved through the detection and matching positioning of sea-crossing bridges. In addition, the monitoring and analysis of moving targets on bridges can be achieved through automatic detection and identification of bridge areas. However, due to the complexity of SAR imaging's characteristics, the coastal terrain distribution, and the marine environment, accurate sea-crossing bridge detection and bridge body recognition are still very difficult.

A bridge is long area spanning over the water. The intersection lines between the bridges and water regions on both sides are parallel, and the distance is usually small. In the SAR images, the single- and double-scattering components of the metal support structures of bridges make them characterized as strong scatterers with high brightness. Most of the existing bridge detection methods are spatial-based methods, in which bridges are detected by the spatial topological relationship between the bridge and the water. Starting from the proximity characteristics of two water regions on both sides of the bridge, References [1–4] extracted the connected sea regions and the water boundaries using different segmentation methods and detected bridges by calculating the distance between different water boundaries. Using the characteristics of the bridge protruding from the water, Chen et al. [5] proposed to extract water regions using a particle filter tracking method and detected bridges by scanning land regions protruding from the water regions. Starting from the feature that the two sides of the bridge are straight and parallel, Yu et al. [6,7] extracted water boundaries by an edge detection method and determined straight parallel lines by the line detection method of the Radon transform. Starting from the high scattering intensity characteristics of bridges, Song et al. [8] detected bridges by extracting the strong scattering regions on the extracted connected water regions using the constant false alarm rate (CFAR) detection method. Due to the complexity of bridge and terrain morphologies, the above existing single-feature-based bridge detection methods are prone to missed detections and false alarms. Thus, Wang et al. [9] integrated the spatial structural features and scattering intensity features of bridges for detection. Liu et al. [10] proposed a bridge detection method based on water network construction using a Markov tree, in which the scattered distributed water branches are connected by the probability graph model of the Markov tree, and bridges were detected by traversing the constructed tree. Chen et al. [11] proposed a deep-learning-based bridge detection method using a multi-resolution attention and balance network. Due to the strong speckle noise of SAR images and the complexity of bridges and surrounding terrains, problems exist in all the above methods. The problem of the spatial-feature-based method is that the water regions are difficult to extract accurately. The problem of the land scanning method is that there are some false alarms formed by natural terrains or raised areas of farmed ponds. The problem of the method based on parallel line detection is that the two sides of some bridges are not parallel. The problems of the CFAR-based method are that only some bridges have high scattering intensity characteristics, and the strong scattering targets and regions on the sea are not only bridges. The problems of deep-learning-based methods are that many training samples are needed to cover bridges with different morphologies and backgrounds.

The sea-crossing bridge is an important class of bridges connecting land regions separated by the sea. The distinctive feature of the sea-crossing bridge is that it usually spans a large sea area, dividing the sea into two parts and appearing as a long, narrow region in the images. The span of the sea-crossing bridge is usually very long, some reaching several thousand meters, and a relatively small width of just a few tens of meters. In SAR images, the large metal supports on both sides of the sea-crossing bridge make its scattering intensity high. In polarimetric SAR images, the single-, double-, and multiple-scattering between the bridge supports, vehicles on the bridge, bridge piers, and the surrounding sea regions make its scattering components very complex. According to the spatial relationship between the bridge and the sea and the geometric features of the bridge, the basic idea of the sea-crossing bridge detection is to segment the sea and land to extract the connected sea regions of a large area first and then determine the close contour parts of the adjacent sea regions or detect the long land regions along the sea. As a summary of the existing bridge detection methods, the difficulties of the sea-crossing bridges' detection mainly include two aspects. One is the accuracies of the sea–land segmentation. Wrong segmentations are prone to occur due to the inherent coherent speckle of SAR images, different sea conditions, and low-scattering soil regions along the coast. The other is the extraction of bridge features. Due to the presence of natural protruding terrains, coastal farming regions, and soils of low scattering, a large number of false alarms will be detected using the spatial-based method.

Because the two sides of the sea-crossing bridge are not straight lines and the intensity distribution of the bridge region is complex, the methods based on parallel line detection and strong scattering region extraction are unable to achieve the detection.

With respect to the sea–land segmentation of polarimetric SAR images, the existing methods mainly use edge-based or region-based active contour models. Sheng et al. [12] proposed a method combining the watershed segmentation algorithm and gradient vector flow snake active contour model. Silveira et al. [13] established the region statistical distribution as a mixed log-normal distribution to obtain the energy term of the level set segmentation. Shu et al. [14] used the narrowband level set segmentation method for the refinement of the sea–land segmentation. Liu et al. [15] proposed a multi-scale level set segmentation method by the analysis of the deviation effect of smoothing to the segmentation. Liu et al. [16] proposed a two-scale active contour model, in which the sea–land is segmented using the region-based and edge-based active contour model in two scale images from coarse to fine, respectively. Modava et al. proposed two methods using the level set segmentation method combining spatial fuzzy clustering [17] and the regional local spectral histogram [18], respectively. Zhu et al. [19] proposed a method by embedding the edge information obtained from edge detection into the geodetic active contour model. The problem of the region-based active contour model is that the sea and land regions are usually not homogeneous. The scattering intensity of some high sea state regions may even be higher than that of some land regions, while the land regions contain man-made buildings, vegetation, soils, and rivers. The distribution of the land and sea regions cannot be fit using the Gamma or Wishart distribution obeyed by homogeneous terrains. Accurate sea–land segmentation is difficult to obtain using the existing region-based segmentation methods based on the homogeneous distribution of the intensity or coherent matrix of a single pixel.

In terms of feature extraction, in addition to the geometric features, the polarimetric features of the sea-crossing bridge are also distinct. Lee et al. analyzed the scattering model and scattering components of the Great Belt Bridge using EmiSAR images in the article [20]. The results showed that the polarization entropy and the scattering angle of the single-, double-, and multiple-scattering regions are much higher than those of the background regions. Apart from this, there is little research work on the analysis of the scattering features of the sea-crossing bridge. However, the polarimetric scattering characteristics of the sea-crossing bridge are similar to other offshore strong scattering targets because the backgrounds of the two targets are the same, and the superstructure of the sea-crossing bridge also forms strong scattering. The difference is that the span of the sea-crossing bridge is large and the superstructure is not dense. Thus, polarimetric parameters used for the detection of offshore targets can be used for the feature extraction of the sea-crossing bridge. In terms of the polarimetric feature extraction of offshore targets, Chen et al. [21] proposed the polarimetric cross-entropy method for the feature extraction of ships. Yang et al. [22] used the polarization similarity parameter [23] to estimate the scattering type and geometric feature of targets first and then proposed a polarimetric feature extraction of ships and man-made buildings based on the generalized optimization of polarimetric contrast enhancement (GOPCE) parameter [24], which enhances the contrast of the coastal terrains and targets with the background by a fusion of the similarity parameter, the polarization entropy, and the power of each channel. Liu et al. [25] also detected offshore oil platforms using the GOPCE features. In addition, the scattering of the sea-crossing bridge is similar to that of some buildings on land. The sea-crossing bridges are buildings with structures such as supports, bridge bodies, piers, and so on. The difference between the two types buildings is that the sea-crossing bridge is a narrow and long region. Thus, the polarimetric parameters used for man-made building extraction can be used for the feature extraction of bridges. Using the reflection symmetry difference between natural features and man-made buildings, Moriyama et al. [26] proposed the polarization correlation coefficient, Ainsworth et al. [27] proposed the circular polarization correlation coefficient, Wang et al. [28] proposed the reflection symmetry parameter,

and Kakimoto et al. [29] proposed the polarization orientation angle parameter for the polarimetric feature extraction of man-made buildings. Liu et al. [30] proposed the ratio parameter of the double-bounce scattering power to the volume scattering power for the recognition of port jetties. By the extraction of polarimetric features, false alarms formed by natural terrains and coastal farming regions can be effectively reduced.

As a summary of the existing bridge detection methods, problems exist in both the sea–land segmentation and feature extractions. In the aspect of the sea–land segmentation, correct results are difficult to obtain in a strong noisy environment. Region-based segmentation methods based on the statistical distribution of a single pixel are prone to obtaining wrong segmentation results because of the possibility of incorrectly connecting low-scattering regions such as soils along the coast or water regions on land with the real sea regions. When the area of the strong-scattering regions is too large, the two-region segmentation method may segment the image into a strong-scattering region and the other. Additionally, the segmented contour of bridges is not smooth due to the coherent speckle noise. Because the scattering intensity of some water regions along the bridge interfered by the double-bounce scattering components between the metal structures of bridges is high, the deviation of the segmented contour is large sometimes. In terms of the feature extraction, there will be many false alarms in the naturally raised regions and the mariculture region along the coast due to the complexity of the terrain topography. Thus, to avoid sea–land segmentation errors and reduce the false alarm rate, we propose a sea-crossing bridge detection method based on windowed level set segmentation and polarization parameter discrimination. To reduce the segmentation errors caused by isolated noisy pixels, the method replaces the single-pixel statistical distribution in the traditional level set segmentation method with a joint statistical distribution of pixel-centered window regions first. The segmented water regions are then merged by fusing the polarization similarity parameters to eliminate some incorrectly segmented water regions. The bridge regions of interest (ROIs) determined by water merging are finally discriminated by the parameters of polarization entropy and scattering angle, effectively reducing the false alarm formed by naturally raised terrains.

The rest of this paper is organized as follows. In Section 2, the proposed method is introduced. In Section 3, experimental results are shown and discussed. The discussion is given in Section 4. The conclusion is given in Section 5.

## 2. The Proposed Method

The diagram of the proposed method is shown in Figure 1, and the proposed method mainly includes three steps. The first step is sea–land segmentation. A windowed level set segmentation method is proposed to carry out the segmentation and obtain the candidate sea regions. The second step is to determine the sea regions related to the sea-crossing bridge. Calculating the distances of contours and polarization similarity among the candidate sea regions, the final sea regions of interest are extracted by merging regions in a fusion of the two distances. The third step is bridge recognition. The regions of interest of the sea-crossing bridge are extracted by determining the land regions between two close sea regions first. The ROIs are then discriminated by the distribution of the polarization entropy and scattering angle parameters of the candidate regions.

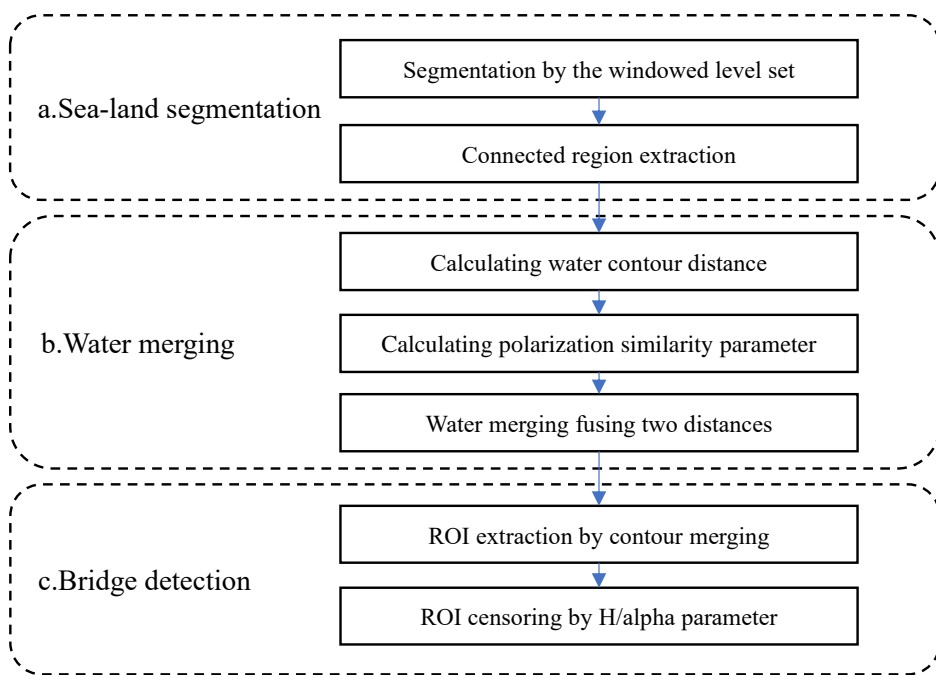

**Figure 1.** Diagram of the proposed method, where ROI denotes region of interest.

### 2.1. Sea–Land Segmentation

In the Bayesian framework, the maximum posterior probability model of the region-based segmentation can be built by modeling the statistical distribution of the sea and land regions. Assuming that the segmentation plane is $\Omega$, $Q(\Omega)$ denotes the segmentation of the image plane, and $I(\Omega)$ is the pixel value of the image, the optimal segmentation is obtained when the segmentation function maximizes the posterior probability $P(Q|I)$ given the observed image $I(\Omega)$. The model is expressed as

$$Q = \underset{Q}{\text{argmax}} \, P(Q|I). \tag{1}$$

The posterior probability $P(Q|I) \propto P(I|Q)P(Q)$, where $P(I|Q)$ is the likelihood function of $Q$ and $P(Q)$ is the prior information of the segmentation. $P(I|Q)$ is determined by the probability distributions of different segmentation regions. $P(Q)$ can be seen as a regularized restriction term about the segmentation curve, which is restricted by the area of the segmented regions and its length:

$$P(Q) \propto \exp\left\{-\gamma_r |D(Q)|^\rho - \lambda_r |\partial Q|\right\}, \tag{2}$$

where $\gamma_r, \rho, \lambda_r$ are constant, $|D(Q)|$ is the area of the segmentation region, and $|\partial Q|$ is the length of the segmentation curve.

The sea–land segmentation is a two-region segmentation problem. If it is assumed that both the sea and land are homogeneous regions, denoted as $R_i(i = 1, 2)$, respectively, and the probability distribution of the coherent matrix $T$ is $f(T|R_i)$, then the likelihood function $P(I|Q)$ can be described as follows:

$$P(I|Q) \propto \prod_i \prod_{(x,y)} f(T(x,y)|R_i), \tag{3}$$

where $(x, y)$ is the two-dimensional coordinate of the pixel.

The maximum posterior probability is equivalent to the minimization of the negative logarithm of the posterior probability. Thus, according to (2) and (3), the energy function of the segmentation can be converted to

$$E(Q) = \gamma_r |D(Q)|^\rho + \lambda_r |\partial Q| - \int_{R_1} \log f(\boldsymbol{T}|R_1) dx dy - \int_{R_2} \log f(\boldsymbol{T}|R_2) dx dy. \tag{4}$$

If the segmentation curve is expressed as $\Gamma$, $R_1$ is the enclosing inner region of $\Gamma$ and $R_2$ is the enclosing outer region of $\Gamma$, then $|D(Q)|$ is the area of $R_1$, and $|D(R_1)|$ and $|\partial Q|$ is the length of the segmentation curve $|\Gamma|$.

The optimal segmentation is obtained by evolving the curve as the gradient of the energy function to the curve until the minimum of the energy function is reached.

### 2.1.1. Level Set Segmentation

The curve evolution of the explicit parametric representation $\Gamma = (x(s), y(s))$ is unable to adapt to changes in the topology of the curve. Thus, the segmentation curve is usually embedded implicitly in a high-dimensional level set function $\Phi(x, y, t)$. The segmentation curve corresponds to the zero level set function $\Gamma = \{(x, y)|\Phi(x, y, t) = 0\}$ in each moment. By evolving the level set function globally, the segmentation curve adapts to the complex topology changes. The energy function is [31]

$$E(\Phi) = \gamma_r \int_R H(\Phi) dx dy + \lambda_r \int_R |\nabla \Phi| dx dy -$$
$$\int_R H(\Phi) \log f(\boldsymbol{T}|R_1) + (1 - H(\Phi)) \log f(\boldsymbol{T}|R_2) dx dy, \tag{5}$$

where $H(\Phi)$ is the step function, if $\Phi \geq 0$, $H(\Phi) = 1$, and if $\Phi < 0$, $H(\Phi) = 0$. $R_1$ corresponds to $\Phi \geq 0$, and $R_2$ corresponds to $\Phi < 0$.

Evolving the level set function along the negative gradient direction, the variational method yields

$$\frac{\partial \Phi}{\partial t} = |\nabla \Phi| \left( \lambda_r \kappa - \gamma_r - \log \frac{f(\boldsymbol{T}|R_2)}{f(\boldsymbol{T}|R_1)} \right), \tag{6}$$

where $\nabla$ is the gradient operator, $\kappa = \text{div}(\nabla \Phi / |\nabla \Phi|)$ is the curvature of $\Phi$, and div(.) denotes the divergence. According to the law of equal perimeters [31], the area regularization term can be restricted by the curve length term, so the term $\gamma_r |\nabla \Phi|$ is usually ignored.

The scattering matrix of each region of the multi-look polarimetric SAR images obeys the complex Wishart distribution. If the average coherent matrix of the region is $\Sigma$ and the number of looks is $L$, then the coherent matrix $\boldsymbol{T} \sim W(\Sigma, L, p)$ is

$$f(\boldsymbol{T}|\Sigma, L, p) = \frac{L^{pL} |\boldsymbol{T}|^{L-p} \exp\left\{ -L \text{tr}\left( \Sigma^{-1} \boldsymbol{T} \right) \right\}}{K(L, p)(|\Sigma|)^L}, \tag{7}$$

where p is the dimension of the Pauli vector, tr(.) denotes the trace of the matrix, $K(L, p) = \pi^{p(p-1)/2} \Gamma(L) \ldots \Gamma(L - p + 1)$, and $\Gamma(.)$ is the Gamma function.

If the average coherent matrix of the region $R_i$ is $\Sigma_i$, the evolution speed of the level set segmentation is obtained by substituting the probability distribution (7) into Equation (6) as:

$$F(\boldsymbol{T}) = \lambda_r \kappa - L\left( \log |\Sigma_1| + \text{tr}\left( \Sigma_1^{-1} \boldsymbol{T} \right) \right) + L\left( \log |\Sigma_2| + \text{tr}\left( \Sigma_2^{-1} \boldsymbol{T} \right) \right). \tag{8}$$

The level set function $\Phi$ is initialized using the signed distance function. Setting the regularization parameter $\lambda_r$, the final segmentation result is obtained by evolving the level set function according to Equation (8) until the energy function is minimized. In the process

of the iterative evolution, the average coherent matrix of the region is estimated by the likelihood estimation method as follows.

$$\widehat{\Sigma}_i = \frac{1}{N_i} \sum_{(x,y) \in R_i} T(x,y). \tag{9}$$

where $N_i$ is the number of pixels in the region $R_i$.

### 2.1.2. Windowed Level Set Segmentation

The problems of the level set segmentation method based on the single-pixel statistical distribution include two aspects. The first is that the segmentation boundaries are not smooth due to the varied scattering of an isolated pixel in the effect of SAR coherent speckle noise. The other is that the missegmentation of several coastline pixels tends to connect the low-scattering regions such as soils along the coast or water regions on land with the sea, leading to incorrect segmentation results. Thus, a joint distribution of a window region centered on each pixel is used for segmentation instead of the isolated pixel distribution.

For pixel $(x, y)$, if the pixel-centered window region of size $w \times w$ is $W$, the joint probability distribution of the window region is proportional to the product of the probability distributions of each pixel:

$$f(W|R_i) \propto \prod_{(u,v) \in W} f(T(u,v)|R_i). \tag{10}$$

Substituting Equation (10) into Equation (5) and adding the normalization factor $1/w^2$, we can obtain the segmentation energy function as

$$E(\Phi) = \lambda_r \int_R |\nabla \Phi| dxdy - \frac{1}{w^2} \int_R H(\Phi) \log f(W|R_1) + (1 - H(\Phi)) \log f(W|R_2) dxdy. \tag{11}$$

From Equation (11), the improved evolutionary speed is

$$\begin{aligned} F(T) =& \lambda_r \kappa - \frac{1}{w^2} \iint_{(u,v) \in W} \log f(T(u,v)|R_1) dudv + \frac{1}{w^2} \iint_{(u,v) \in W} \log f(T(u,v)|R_2) dudv \\ =& \lambda_r \kappa - L\left(\log|\Sigma_1| + \mathrm{tr}\left(\Sigma_1^{-1} \bar{T}\right)\right) + L\left(\log|\Sigma_2| + \mathrm{tr}\left(\Sigma_2^{-1} \bar{T}\right)\right). \end{aligned} \tag{12}$$

where $\bar{T}$ is the average coherent matrix of a window region.

Setting the initial level set function and the regularization parameter, the final segmentation result can be obtained by evolving the level set function using (12) until convergence. The estimation of the average coherent matrix of each segmentation region during the iterative process is also determined by (9).

### 2.2. Water Merging

Due to the presence of water regions on the land and strong scatterers on the sea, the segmentation results obtained by the level set segmentation method contain several land regions on the sea and several water regions on the land. Thus, to extract the sea regions related with the sea-crossing bridge, the segmented water regions need to be censored by their areas and merged by distance.

### 2.2.1. The Water Merging Algorithm

If the segmentation result is $A$, the set of connected water regions extracted by the segmentation result is $B = \{b_k\}(k \in \{1, \ldots, N_b\})$. The set is $C = \{c_l\}(l \in \{1, \ldots, N_c\})$ after censoring by the area thresholding process of $Area_{th}$. The contour boundary of each water obtained by the boundary tracking algorithm is $Z = \{z_l\}(l \in \{1, \ldots, N_c\})$, where $z_l$ denotes the set of boundary points of the water region $l$. The contour distance matrix between each of the two water regions is $D = \{d_{ij}\}$. Because the width range of bridges is known, two water regions can be merged into the same water region if the contour distance

is less than the maximum of the bridge width. Thus, if the distance threshold is set to $D_{th}$, the water region merging algorithm is as follows (Algorithm 1).

---

**Algorithm 1** Region merging algorithm.

---

**Input:**
      Water regions: $C$;
      The contour distance matrix: $D$;
      The contours of water regions: $Z$;
      A flag array for all water regions to be merged: $F$;
      The size of $C$: $N_c$;
      A queue for the storage of unmerged water regions: $Q$.

**Initialization:**
      initialize all elements in $F$ to zero;
      Find the water with maximum area $c_m$, $Q.push(m)$.

**Region_merge(region $C$, dmatrix $D$, contour $Z$, flag $F$, size $N_c$, queue $Q$)**

 1: **while** $Q.size() > 0$ **do**
 2:   $Seed = Q.front()$
      $F(Seed) = true$
      $Q.pop()$
 3:   **for** $i = 1$ to $N_c$ **do**
 4:     **if** $d_{seed,i} < D_{th}$ **and** $!F(i)$ **then**
 5:       $Q.push(i)$
 6:     **end if**
 7:   **end for**
 8: **end while**

**State:** $Q.size()$, $Q.front()$, $Q.pop()$, and $Q.push(i)$ denote the current size, the front element, the pop process, and the push process of the queue; $d_{seed,i}$ denotes the minimum contour distance of water regions' seed and $i$.

---

All the water regions $G = \{g_k\}(k \in \{1, \ldots, N_g\})$ satisfying the flag of $F(g_k)$ being true are determined as the final water segmentation result.

2.2.2. Water Merging Algorithm Fusing Polarization Similarity Parameter

Because of the presence of low-scattering regions such as coastal soil and farming regions, some errors may exist in the segmentation results using Algorithm 1. The cause is that only the contour distance is used in the merging process. Considering that the polarimetric scattering characteristics between these erroneous water regions and the real water regions are different, the polarization similarity parameter among the suspected water regions is used for the improvement of the water merging algorithm.

For any given two regions $X$ and $Y$, if their average coherent matrices are $T_X$ and $T_Y$, respectively, the polarization similarity parameter [23] of the two regions is defined as

$$r(T_X, T_Y) = \frac{\langle T_X^0, T_Y^0 \rangle}{\|T_X^0\|_F \|T_Y^0\|_F} = \frac{\left| \text{tr}\left( (T_X^0)^H T_Y^0 \right) \right|}{\sqrt{\text{tr}\left( (T_X^0)^H T_X^0 \right) \text{tr}\left( (T_Y^0)^H T_Y^0 \right)}}. \tag{13}$$

where $T^0 = De(T)$ denotes the deorientation operation, $\langle \cdot, \cdot \rangle$ denotes the inner product operation, and $\|\cdot\|_F$ denotes the matrix Frobenius norm.

Considering that the orientation angle of the water is almost negligible and the incorrectly segmented regions may have a certain orientation angle, the deorientation operation is meaningful. For the candidate water set $\{c_l\}$, the average coherent matrix $T_l$ is calculated first. The polarization similarity parameters of each of the two regions are then calculated separately according to Equation (13) to obtain the similarity parameter measure matrix $\rho = \{\rho_{ij}\}$. The values of the polarization similarity parameter of the two regions are in the

range $0 \sim 1$. Statistical results show that, if the polarization similarity parameter of the two regions is higher than 0.9, the two regions can be considered as a homogeneous region. Thus, the threshold of the polarization similarity parameter $\rho_{th}$ is set to 0.9 or higher.

According to the similarity parameter measure matrix and the similarity threshold, the water merging algorithm fusing polarization similarity parameter can be obtained by modifying the merging constraint in Algorithm 1 from $d_{seed,i} < D_{th}$ to $d_{seed,i} < D_{th}$ and $\rho_{seed,i} < \rho_{th}$, where $\rho_{seed,i}$ is the polarization similarity parameter between the water regions' seed and $i$. In practice, there may be several sea regions separated by large land regions instead of the sea-crossing bridges. However, only the sea region of the largest area can be extracted by Algorithm 1. Thus, it is necessary to improve Algorithm 1 to be applicable in the case of water merging containing multiple subsets of the sea. By marking all the major water regions using a flag array $MF$, the multi-region merging algorithm can be described as follows (Algorithm 2).

---

**Algorithm 2** Multi-region merging algorithm.

---

**Input:**
       Water regions: $C$;
       The contour distance matrix: $D$;
       The contours of water regions: $Z$;
       A flag array for all water regions to be merged: $F$;
       The size of $C$: $N_c$;
       A queue for the storage of unmerged water regions: $Q$.

**Initialization:**
       initialize all elements in $F$ to zero;
       set an area threshold $Area_{th2}$ for the major water regions' determination;
       initialize a flag array $MF$ to zeros for the marking of all major waters.

**Multi_region_merge(region $C$, dmatrix $D$, contour $Z$, flag $F$, size $N_c$, queue $Q$)**

1:  **for** $i = 1$ to $N_c$ **do**
2:    **if** $C(i).area() > Area_{th2}$ **then**
3:      $MF(i) = true$
4:    **end if**
5:  **end for**
6:  **while** $\forall i \; \exists MF(i)$ is true and $F(i)$ is false **do**
7:    **for all** $i = 1$ to $N_c$ such that $MF(i)$ is true and $F(i)$ is false **do**
8:      $m \leftarrow max(C(i).area())$
9:    **end for**
10:   $Q.push(m)$
11:   **while** $Q.size() > 0$ **do**
12:     $Seed = Q.front()$
       $F(Seed) = true$
       $Q.pop()$
13:     **for** $i = 1$ to $N_c$ **do**
14:       **if** $d_{seed,i} < D_{th}$ **and** $\rho_{seed,i} < \rho_{th}$ **and** $!F(i)$ **then**
15:         $Q.push(i)$
16:       **end if**
17:     **end for**
18:   **end while**
19:  **end while**

---

### 2.3. Bridge Detection

2.3.1. Extraction of Bridge Regions of Interest

According to the spatial relationship between sea-crossing bridges and the sea, bridge regions of interest can be determined by extracting the proximity contours of each of two adjacent water regions based on the sea segmentation results. The process of the adjacent

water region extraction is similar to that of the water merging. However, because a water region may be adjacent to more than one water region, the algorithm for the ROI extraction is as follows (Algorithm 3).

---

**Algorithm 3** Bridge ROI extraction algorithm.

---

**Input:**
  Water regions extracted by Algorithm 2: $G$;
  The contour distance matrix: $D$;
  The contours of water regions: $Z$;
  The size of $G$: $N_g$.
**Output:**
  Water regions extracted by Algorithm 3: $ROI$.

$ROI$ = **Bridge_ROI(region** $G$, **dmatrix** $D$, **contour** $Z$, **size** $N_g$)

1:  **for** $i = 1$ to $N_g$ **do**
2:   **for** $j = i + 1$ to $N_g$ **do**
3:    **if** $d_{i,j} < D_{th}$ **then**
4:     $ROI \leftarrow$ *all points on* $z_i$ *and* $z_j$ *s.t.* $dist(z_i, z_j) < D_{th}$
5:    **end if**
6:   **end for**
7:  **end for**
8:  $ROI \leftarrow$ *all land pixels in the minimum circumscribed rectangular ROI*
9:  **return** $ROI$

---

### 2.3.2. ROI Censoring by Polarization Parameters

Because of the complexity of coastal terrain and the presence of farming dams, some false alarms occur in the detection method of the water merging by distance using Algorithm 3. To reduce the false alarm, polarimetric scattering parameters need to be extracted. Due to the double-bounce and multiple scattering components between the metal structure and the bridge deck, the polarimetric scattering components of the real sea-crossing bridge are more complex than that of the false alarm. Lee analyzed the scattering characteristics of the Great Belt Bridge in EMISAR in [20]. The statistical results showed that the average scattering angle parameters [32] of the single-, double-bounce, and multiple-scattering regions of the bridge were much higher than those of other regions. An example statistical result is shown in Figure 2 using TerraSAR-X data in San Francisco. Figure 2a shows the Pauli pseudo-color image, where the region marked in red is a selected coastal natural raised terrain. Figure 2b shows the distribution of the polarization entropy and scattering angle within the region of the Richmond Bridge, where the gray region is the distribution of the whole image, the red line is the contour line of the overall distribution, and the blue dots are the scatter distribution of the pixels in the bridge region. Figure 2b shows that the scattering angles of the bridge region are all distributed in the region above 45°, and the polarization entropy is mostly distributed in the region above 0.5. Figure 2c shows the $H/\alpha$ distribution of the coastal natural raised terrain. We can see that the alpha angles of most pixels are below 45°. Figure 2d shows the alpha angle histograms of the two regions, where the red color is the histogram of the bridge and the green color is the coastal raised terrain region. It can be observed that the difference of the alpha angle distributions between the sea-crossing bridge and the false alarm region is large. Thus, the detected ROIs can be discriminated by the $H/\alpha$ distribution. According to the statistical results, if the proportion of pixels of $H > 0.5$ and $\alpha > 45°$ is higher than a threshold $\sigma$, the ROI can be identified as the sea-crossing bridge.

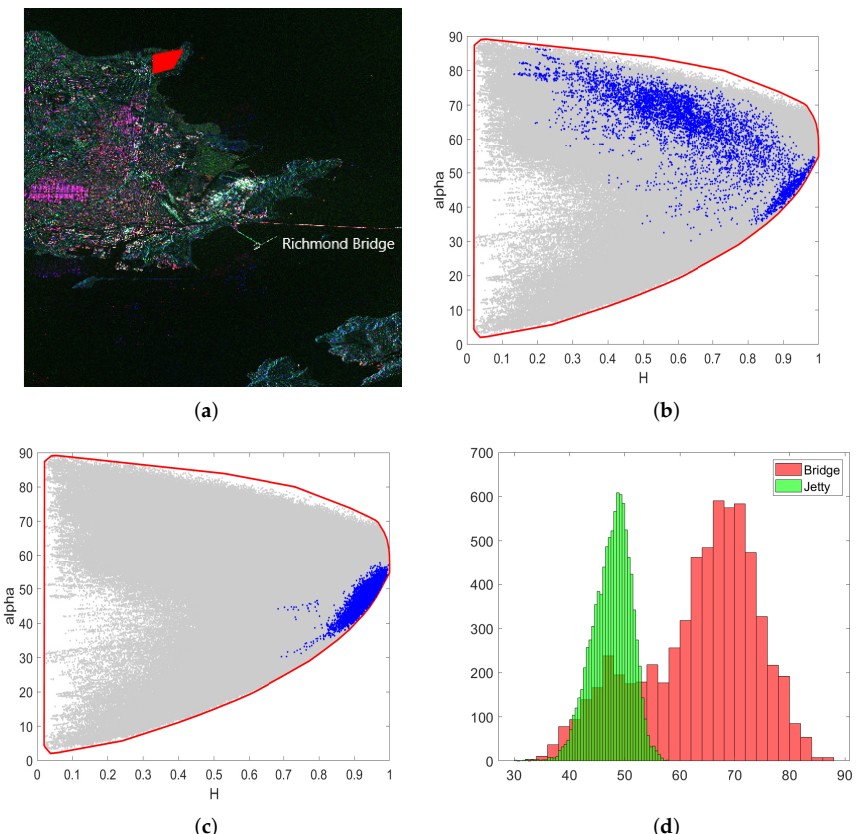

**Figure 2.** Distribution of polarization entropy and scattering angle of the sea-crossing Richmond Bridge in the San Francisco region from TerraSAR-X. (**a**) The Pauli pseudo-color image of the sample data, where the region marked in red is a selected coastal natural raised terrain. (**b**) Results of the sea-crossing bridge, where the gray region is the distribution of the whole image, the red line is the contour line of the overall distribution, and the blue points are the scatter distribution of the pixels in the bridge region. (**c**) Results of the raised terrain region along the coast. (**d**) Comparison of the scattering angle histograms of the two regions, where the red area is the bridge region and the green area is the coastal raised terrain region.

## 3. Experimental Results and Analysis

Multiple AirSAR, RADARSAT-2, and TerraSAR-X single-look and multi-look quad-polarization data from the San Francisco region of the United States, the Singapore coastal region, the Fuzhou region of Fujian province, and the Zhanjiang region of Guangdong province, China, which contain sea-crossing bridges, were selected for the experiment. The detail information of the data is shown in Table 1. Except for the AirSAR San Francisco data with a size of 900 × 1024 and a resolution of 12 m × 6 m, all other data are close to 6000 × 3000 in size, RADARSAT-2 data with a resolution close to 5 m × 5 m, and TerraSAR-X data with a resolution close to 2 m × 6.5 m. Figure 3 shows the Pauli pseudo-color images of different data, where "R", "G", and "B" denote the three components of the Pauli vector. To keep the balance of color contrast, each component is divided by its mean when the image is shown. Because the acquisition track of the testing TerraSAR-X data is different from the AirSAR and RADARSAT-2 data, the figures of Data 6 and Data 7 in Figure 3 are back-side compared with the corresponding Google map. From Figure 3, we can observe that the sea-crossing bridge spans over the sea with different topographic distributions. Because the morphology of the bridge and the surrounding environment are varied, the scattering components of the sea-crossing bridge are varied. Due to the varied-single, double-bounce, and multiple-scattering components of the bridge superstructure, the scattering intensity of the bridge varies and the two sides of the bridge are not parallel. The Golden Gate Bridge and Richmond Bridge in the San Francisco region are brighter

than the bridges in other regions due to their wide width and more complex structural components. Due to the simultaneous inclusion of three bridges, the width of the Golden Gate Bridge is greater than that of the other bridges. Unlike the other bridges that appear as a narrow strip region, the Richmond Bridge has a large span and the sides are not straight.

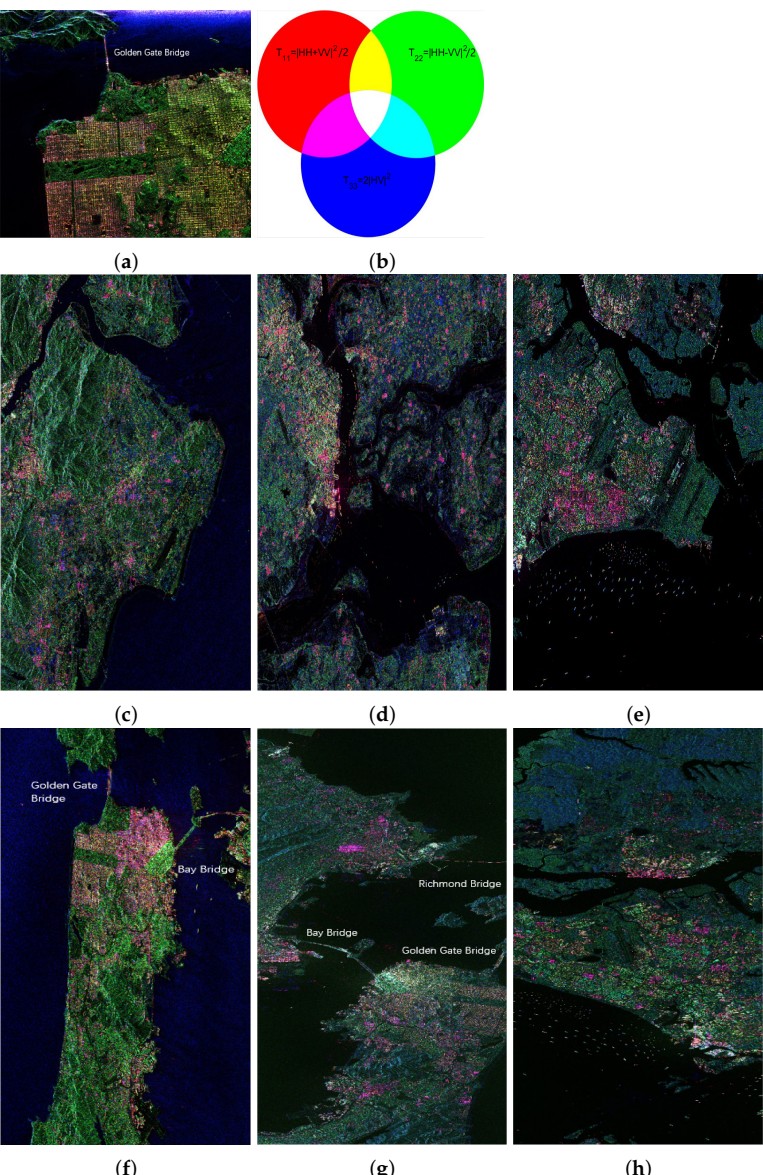

**Figure 3.** The Pauli pseudo-color images of the experimental data. (**a**) AirSAR San Francisco. (**b**) The color code of the Pauli image, where "R", "G", and "B" denote the three components of the Pauli vector, respectively, and $(HH, VV, HV)$ are the three components of the Sinclair matrix. (**c**) RADARSAT-2 Fuzhou. (**d**) RADARSAT-2 Zhanjiang. (**e**) RADARSAT-2 Singapore. (**f**) RADARSAT-2 San Francisco. (**g**) TerraSAR-X San Francisco. (**h**) TerraSAR-X Singapore.

**Table 1.** Details of the experimental data, where **UTC** stands for the international standard time, **AOI** stands for the angle of incident, $m \times m$ stands for meter times meter, MLC stands for multi-look complex data, and SLC stands for single-look complex data.

| Data | Scene | Sensor | Band | Mode | Size | Resolution ($m \times m$) | UTC | AOI (º) |
|------|-------|--------|------|------|------|---------------------------|-----|---------|
| 1 | San Francisco | AirSAR | L | MLC | $900 \times 1024$ | $12 \times 6$ | 1992 | 47.3 |
| 2 | Fuzhou | RADARSAT-2 | C | SLC | $6140 \times 3332$ | $4.73 \times 4.81$ | 10/20/2013 10:05:53 | 36.02 |
| 3 | Zhanjiang | RADARSAT-2 | C | SLC | $5937 \times 3920$ | $4.73 \times 4.95$ | 03/23/2012 22:28:47 | 43.31 |
| 4 | Singapore | RADARSAT-2 | C | SLC | $6161 \times 4256$ | $4.73 \times 4.80$ | 19/01/2013 11:31:08 | 47.4 |
| 5 | San Francisco | RADARSAT-2 | C | SLC | $6000 \times 2800$ | $4.73 \times 4.82$ | 09/04/2008 02:01:33 | 28.9 |
| 6 | San Francisco | TerraSAR-X | X | MLC | $5800 \times 3000$ | $1.84 \times 6.59$ | 03/10/2014 11:05:42 | 39.7 |
| 7 | Singapore | TerraSAR-X | X | MLC | $5500 \times 2500$ | $2.06 \times 6.59$ | 10/03/2014 11:07:06 | 34.7 |

Referring to Google Earth for the ground-truth mapping, the performance was evaluated using the detection rate, false alarm rate, Intersection over Union (IoU) and Intersection over Ground-truth (IoG) indices. If DR denotes the recognized bridge region and GT denotes the labeled real bridge region, the IoU index is defined as the ratio of the intersection of DR and GT to the union of both. Considering that the main concern is usually the proportion of the real region of the bridge that is correctly detected in practice, the IoG metric is defined as the ratio of the intersection of DR and GT with GT.

$$
IoU = \frac{DR \cap GT}{DR \cup GT}
$$
$$
IoG = \frac{DR \cap GT}{GT}
$$

(14)

where $\cap$ denotes the intersection operation and $\cup$ denotes the union operation.

Five experiments were carried out to evaluate the performance of the proposed method. The algorithm processes are shown in detail using the AirSAR data of San Francisco in Experiment 1. Experiment 2 evaluated the detection performance in different regions. Experiment 3 compared the detection performance using datasets of the same region in different bands. Experiment 4 compared the performance of the algorithm under different parameters. The performance of the windowed level set segmentation and the final detection performance were compared under different window sizes. Experiment 5 compared the performance differences between the proposed method and the bridge detection method based on the spatial structure.

*3.1. Parameters Setting*

When the windowed level set segmentation method was carried out, the curve regularization parameter $\lambda_r$ was set to 0.2 and the window size $w$ was set to 5, which was determined by comparing the performances under different parameters in Experiment 4. When performing water merging, if the image resolution is $R_x \times R_y$, the maximum width of the bridge is $W_b$, and supposing the span of the sea-crossing bridge is at least 1 km, the area threshold $Area_{th}$ was set to $10^3/R_x \times 10^3/R_y$ and the distance threshold $D_{th}$ was set to $W_b/\sqrt{R_x^2 + R_y^2}$. When performing bridge recognition, the high $H/\alpha$ area ratio threshold $\sigma$ was set to 1/4 since there was no bracket part above most of the bridge region.

*3.2. Example Results*

In the first experiment, the procedure of the proposed method was demonstrated using the AirSAR San Francisco data. The experimental results are shown in Figure 4. Figure 4a shows the Pauli pseudo-color image of the data and the ground-truth of the bridges, where the green pixels in the red boxes are the marked bridge region. We can find that there exists only one sea-crossing bridge, the Golden Gate Bridge. The sea regions in the figure are presented as different sea states with different colors, while the land includes mountainous,

urban, vegetation, and soil regions. It is difficult to segment the sea and land accurately using the two-region-based segmentation method. The result of sea–land segmentation using the windowed level set method is shown in Figure 4b. It can be observed that the high sea state region in the upper right corner was incorrectly segmented as the land region, while part of the land region along the mountainous coast was incorrectly segmented as the sea region. Because the scattering components of the sea-crossing bridge are complex, part of the water near the bridge was incorrectly segmented. Because the segmentation contours of the sides of the bridge were not smooth, the geometric features of the bridge were difficult to extract. Figure 4c shows the result of sea and land segmentation after the post-processing of area thresholding. Through the area thresholding, most of the small water and land regions that are not relevant to the sea-crossing bridge were eliminated. Figure 4d shows the result of sea–land segmentation after water merging, where the red pixels are the recognized ROI. We can observe that almost all the bridge pixels were correctly extracted. Figure 4e shows the high entropy and high scattering angle regions by the $H/\alpha$ thresholding segmentation, where the recognized ROI is marked in a red box. It can be observed that most pixels of the bridge body are in the high-entropy and high-scattering angle regions. We also can see that the wrong water and land segmentation region in the upper right corner of Figure 4d can be corrected by the $H/\alpha$ segmentation. For the proposed method, only the recognized ROI needs to perform the $H/\alpha$ computation to discriminate whether it is a false alarm or not. Figure 4f shows the final bridge detection result, where the red pixels are the detected bridge and the green pixels are the ground-truth. Comparing the two regions, we can find that the recognized bridge region almost overlaps with the ground-truth except for some pixels near the bridge outline. In the following experiments, the color markings in the experimental results were consistent with those in Experiment 1 and will not be repeated here.

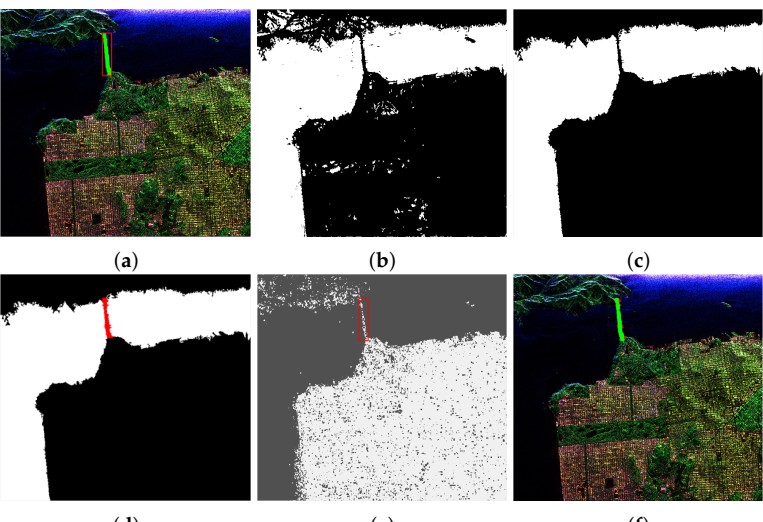

**Figure 4.** Sea-crossing bridge detection results of the data of AirSAR San Francisco. (**a**) Pauli pseudo-color image and ground-truth of bridges, where green pixels in red boxes are the marked bridge region. (**b**) The result of sea–land segmentation using the windowed level set method. (**c**) The result of sea–land segmentation after the post-processing of area thresholding. (**d**) The result of sea–land segmentation after water merging, where the red pixels are the recognized ROI. (**e**) The high-entropy and high-scattering angle regions by the $H/\alpha$ thresholding segmentation, where the recognized ROI is marked in a red box. (**f**) The final bridge detection result, where the red pixels are the detected bridge and the green pixels are the ground-truth.

### 3.3. Performance Comparison under Different Regions

To verify the applicability and robustness of the proposed method in different scenarios, experiments were carried out using RADARSAT-2 polarimetric SAR data (Data 2–4 in Table 1) in the Fuzhou, Fujian, Zhanjiang, Guangdong, and Singapore regions. The experimental results are shown in Figure 5, where Figure 5a1–a3 are Pauli pseudo-color images. The observation shows that one sea-crossing bridge is distributed in each of the Fuzhou and Singapore region data, and two sea-crossing bridges are distributed in the Zhanjiang region data (there is another bridge at the left border of the Fuzhou and Zhanjiang region data, but it was ignored because it is too close to the border and can no longer be distinguished). Both the Fuzhou and Zhanjiang data have large areas of beach along the coast, and the Zhanjiang and Singapore data have a large number of strongly scattered targets at sea, all of which increase the difficulties for accurate sea–land segmentation, while the low-scattering areas and the farming dams along the coast are prone to forming false alarms. Figure 5b1–b3 show the results of the sea–land segmentation. We can find that there are many small regions in the sea and land segmentation results. There are several mesh regions along the coast, in which the long dams across the water easily form false alarms. Figure 5c1–c3 show the results of water merging, where the red pixels are the bridge ROIs detected by the merging process. From Figure 5c1,c2, false alarms are found in both Fuzhou and Zhanjiang data. Comparing the map, the false alarm of Fuzhou data is the coastal port breakwater. The false alarm was caused by the wrong segmentation of the coastal low scattering region into the sea, resulting in the formation of an area similar to the sea-crossing bridge. Figure 5d1–d3 show the results of the recognized bridges after censoring by the thresholding of $H/\alpha$. Figure 5e1–e3 show the enlarged results of the area near the sea-crossing bridge. Observing Figure 5d1–d3, it can be seen that the sea-crossing bridges are correctly recognized in each image. Because of the difference of $H/\alpha$ between the false alarms and the sea-crossing bridges, the false alarms detected by water merging were correctly eliminated. Observing Figure 5e1–e3, we can see that the detected bridge regions have small differences from the real regions, and the main differences are located at the two ends of the bridges. Because some land pixels near the ends of the bridge are too close to the water boundaries, there were some false pixels in the ends of the bridge under the set distance threshold.

Table 2 shows the detection performance of the three datasets. All four sea-crossing bridges were detected correctly in the three datasets. Because the span of the bridge is relatively smaller and the background is better than those of the other two data, the IoG and IoU indices of Data 4 were better than the other two data, reaching 90.37 and 78.56, respectively. Because Data 3 contains two bridges with complex scattering components on both sides of the bridge, more pixels in the sea were wrongly segmented into the bridge region. Thus, the IoG and IoU indices were lower than those of the other two data, at 82.18 and 69.58, respectively.

**Table 2.** Performance of RADARSAT-2 polarimetric SAR data for sea-crossing bridge detection in Fuzhou, Fujian, Zhanjiang, Guangdong, and Singapore regions, where Num denotes the number of targets. Pd denotes the detection rate. Pf denotes the false alarm rate. IoG denotes the intersection over ground-truth. IoU denotes the intersection over union.

| Data | Num | Correct | Pd (%) | False Alarm | Pf (%) | IoG (%) | IoU (%) |
|------|-----|---------|--------|-------------|--------|---------|---------|
| 2 | 1 | 1 | 100 | 0 | 0 | 88.05 | 71.86 |
| 3 | 2 | 2 | 100 | 0 | 0 | 82.18 | 69.58 |
| 4 | 1 | 1 | 100 | 0 | 0 | 90.37 | 78.56 |

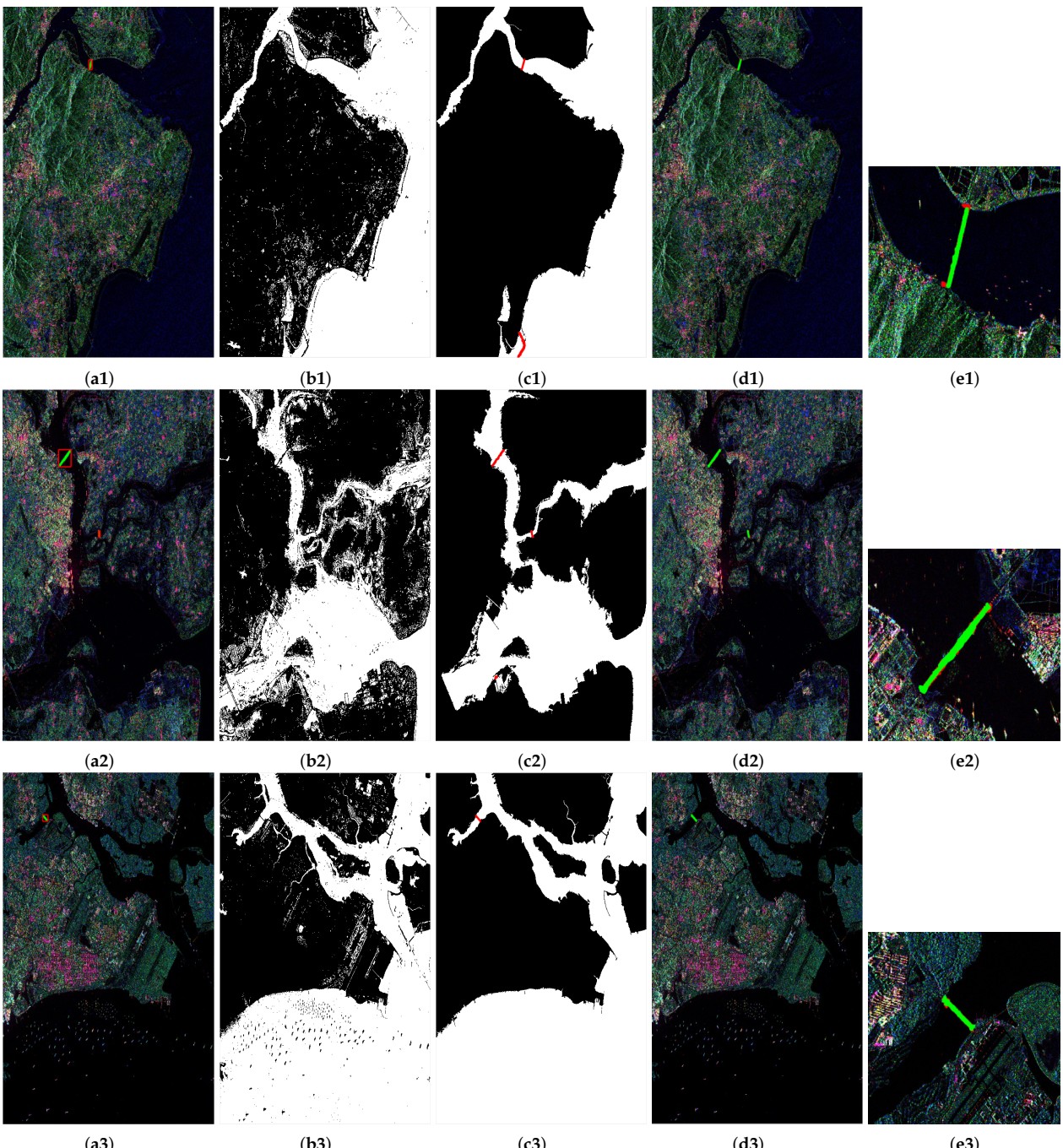

**Figure 5.** Comparison of detection results of sea-crossing bridges under different scenarios. ai-ei are the results of RADARSAT-2 polarimetric SAR data in Fuzhou, Fujian, Zhanjiang, Guangdong, and Singapore, respectively. (**a1**–**a3**) The Pauli pseudo-color images and the ground-truth of bridges. (**b1**–**b3**) Sea–land segmentation results. (**c1**–**c3**) Water merging results, where the red pixels are the bridge regions detected by Algorithm 3. (**d1**–**d3**) The bridge results recognized by the censoring of $H/\alpha$. (**e1**–**e3**) The enlarged results of the region near the sea-crossing bridges.

### 3.4. Performance Comparison under Different Bands

To verify the robustness of the proposed method under different bands, experiments were carried out using AirSAR, RADARSAT-2 and TerraSAR data in the San Francisco region. From Figure 3, we can see that three sea-crossing bridges, the Golden Gate Bridge, Richmond Bridge, and Bay Bridge, are distributed in the region. The results are shown in Figure 6, where Figure 6a1–a3 show the Pauli pseudo-color image and the ground-truth of

the bridges. We can find that the distributions of the scattering intensities of the sea and bridges in the three bands are different. The scattering intensity of the sea in the X-band is higher than that in the C-band and L-band. The scattering intensity of the urban region near the coast of San Francisco is higher. Because the range resolution of TerraSAR-X is higher than that of RADARSAT-2 and AirSAR, the width of the Golden Gate Bridge is wider than that of the other two images. Figure 6b1–b3 show the results of sea–land segmentation. We can observe that a large number of sea pixels near the Richmond Bridge in the C-band data were incorrectly segmented as land pixels. Figure 6c1–c3 show the results of the bridge detection, where the enlarged region near the Golden Gate Bridge is shown in Figure 6d1–d3. We can see that there were some false alarm pixels at both ends and sides of the detected bridges. The Richmond Bridge had fewer false alarm pixels in the near straight half and more in the curved half due to the large span, the curved bridge, and the connection to an island in the middle.

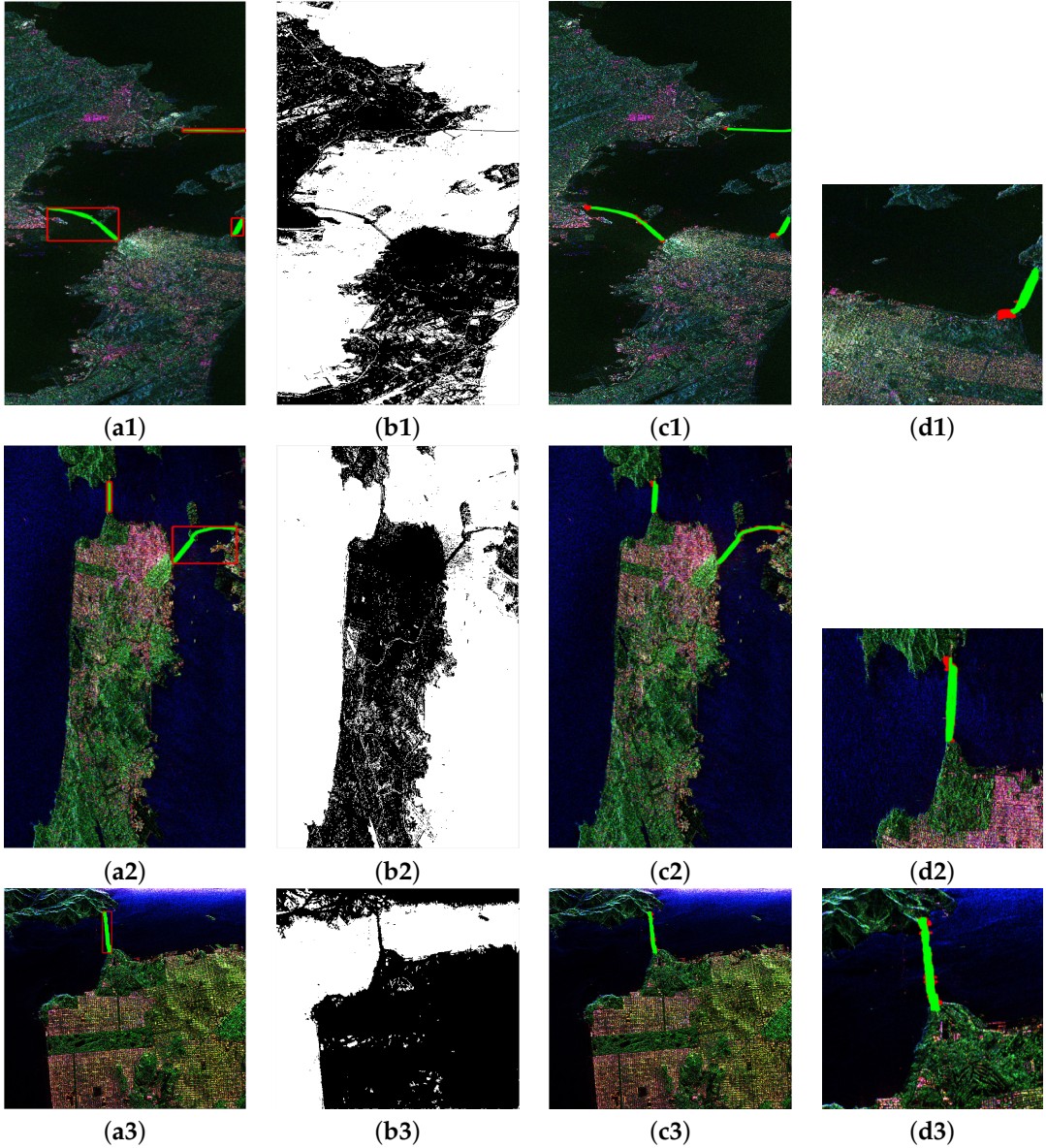

**Figure 6.** Detection results of sea-crossing bridges with different bands of polarimetric SAR data in San Francisco. (**a1**–**a3**) The Pauli pseudo-color images and the ground-truth of bridges. (**b1**–**b3**) Sea–land segmentation results. (**c1**–**c3**) The bridge detection results. (**d1**–**d3**) The enlarged results of the region near the Golden Gate Bridge. (**a1**–**d1**) Results of TerraSAR-X data. (**a2**–**d2**) Results of RADARSAT-2 data. (**a3**–**d3**) Results of AirSAR data.

The detection performance of each bridge in different data are listed in Table 3. It can be observed that the AirSAR data had a higher mIoU index (IoU index of the whole image) of 74.8 than the other two data due to the inclusion of only the Golden Gate Bridge. Because the percentage of the bridge region being incorrectly segmented into sea pixels, the RADARSAT-2 data had the highest mIoG index (IoG index of the whole image) of 88.94. For the Golden Gate Bridge, which exists in all three bands, the IoU index of the AirSAR data was better than that of the other two bands, and the IoG index of the RADARSAT-2 data was better than that of the other two bands. For the Bay Bridge, the IoG performances of the TerraSAR-X and RADARSAT-2 data were comparable, but the IoU index of the RADARSAT-2 data was superior. For the Richmond Bridge with the TerraSAR-X data, the IoU index was better than that of the Golden Gate and Bay Bridges due to the narrow bridge and fewer false alarm pixels.

**Table 3.** Detection performances of each bridge in the San Francisco region under different bands, where **mIoG** denotes the average intersection over ground-truth value. **mIoU** denotes the average intersection over union.

| Sensor | Bridge | IoG (%) | IoU (%) | mIoG (%) | mIoU (%) |
|---|---|---|---|---|---|
| | Golden Gate Bridge | 75.62 | 52.50 | | |
| TerraSAR-X | Bay Bridge | 89.38 | 59.38 | 80.66 | 60.13 |
| | Richmond Bridge | 77.00 | 68.51 | | |
| RADARSAT-2 | Golden Gate Bridge | 90.07 | 67.30 | 88.94 | 65.94 |
| | Bay Bridge | 87.81 | 64.58 | | |
| AirSAR | Golden Gate Bridge | 85.24 | 74.80 | 85.24 | 74.80 |

*3.5. Performance Comparison under Different Parameters*

To verify the effectiveness of the proposed method, the segmentation and detection performances were compared under different segmentation window sizes. Figure 7 shows the detection results of the TerraSAR-X San Francisco data when the windows were set to 1, 3, 5, 7, and 9, respectively. Figure 7a1–a5 show the segmentation results of the level set. Figure 7b1–b5 show the segmentation results and the extraction results of the ROI of the bridge body after the water merging process. Figure 7a*i*–c*i* show the detection results when the window sizes are 1, 3, 5, 7, and 9, respectively. Comparing the results of Figure 7b1–b5, with the increase of the window size, the isolated pixels of the land internally segmented as water were significantly reduced, and the segmentation results of the land region along the coast were more connected. When the window size was one, the image was incorrectly segmented into the strong scattering region and other regions from Figure 7a1, which led to the inability of the two region segmentation methods to correctly segment the sea and land from Figure 7b1. When the window size was three, some pixels in the coastal low-scattering region were incorrectly segmented as the sea surface pixels, resulting in the internal low-scattering region connected to the sea region and leading to the incorrect segmentation of the sea region in the lower right corner and the formation of a false alarm target from Figure 7b2. However, when the window size was larger than five, more water pixels along the sea-crossing bridges were wrongly segmented as land regions due to the smoothing effect. Because the wider land leads to the contour distance of two adjacent water regions related to the Gold Gate Bridge being larger than the distance threshold $D_{th}$, only part of the bridge is detected from Figure 7b4,b5. As shown in Figure 7c1–c5 for the bridge detection results, the segmentation result of a window size of five is more favorable for the detection of the sea-crossing bridge. As listed in Table 4 for the detection performance under different window sizes, the accuracy rate increased with the increase of the window size when the window size was smaller than five. The bridge failed to be detected in the case of a window size of one. The IoG index of a window size of five was 7.4% higher that of a window size of three, and the IoU index was 9.8% higher. The accuracy rate decreased with the increase of window size when the window size was larger

than five. The IoG index of a window size of five was 5.11% higher that of a window size of seven, and the IoU index was 2.75% higher.

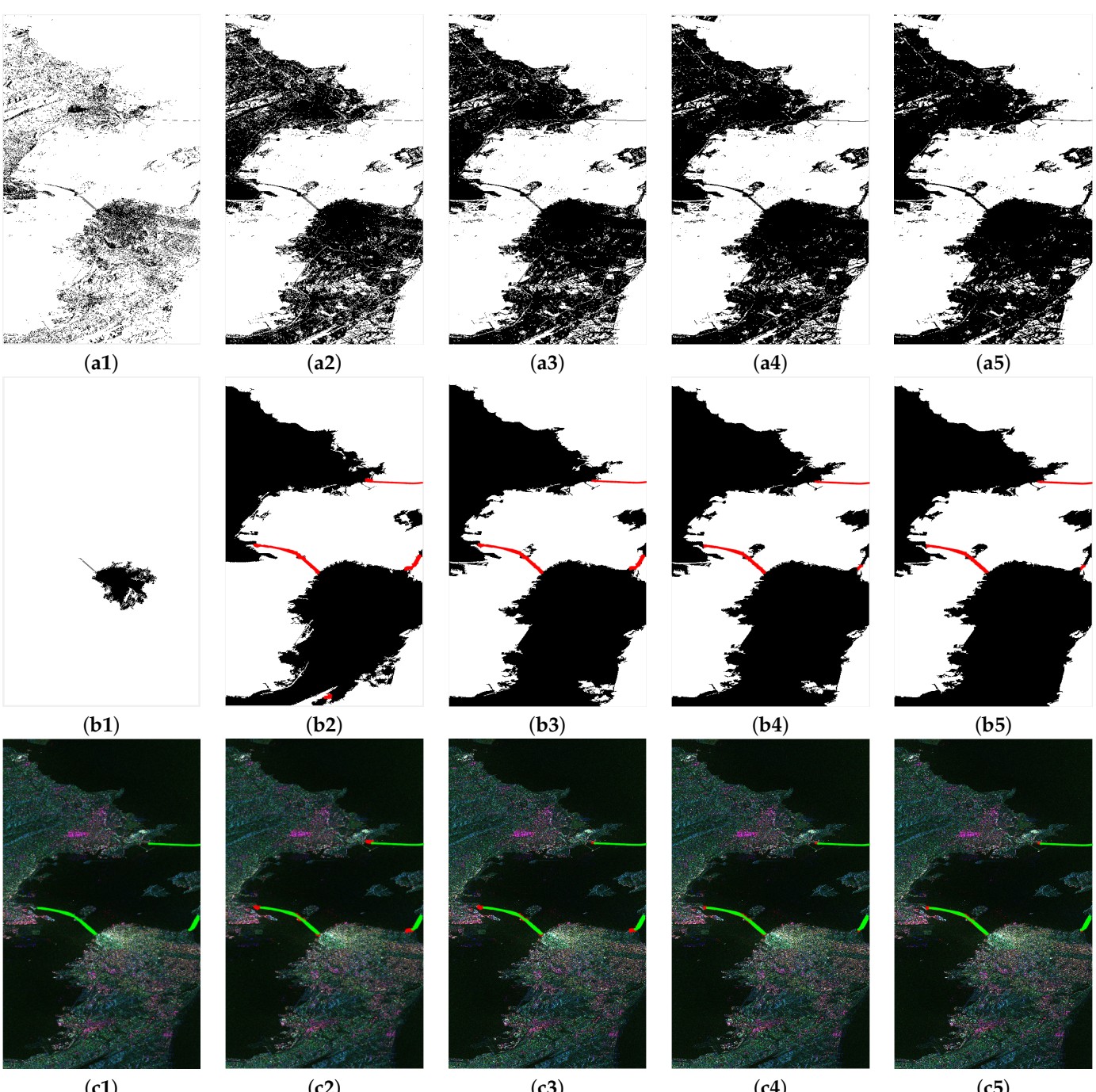

**Figure 7.** Performance comparison of the TerraSAR-X San Francisco data using the windowed level set segmentation under different window sizes. (**a1**–**a5**) The level set segmentation results. (**b1**–**b5**) The segmentation results after water merging and the detection results of the region of interest of the bridge. (**c1**–**c5**) The results of bridge recognition. (**a***i*–**c***i*) The detection results when the window sizes are 1, 3, 5, 7, and 9, respectively.

**Table 4.** Comparison of the detection performance of the TerraSAR-X San Francisco data using the windowed level set segmentation under different window sizes.

| Win | Num | Correct | Pd (%) | False Alarm | Pf (%) | IoG (%) | IoU (%) |
|---|---|---|---|---|---|---|---|
| 1 | 3 | 0 | 0 | 0 | 0 | 0 | 0 |
| 3 | 3 | 3 | 100 | 0 | 0 | 73.27 | 50.30 |
| 5 | 3 | 3 | 100 | 0 | 0 | 80.66 | 60.13 |
| 7 | 3 | 3 | 100 | 0 | 0 | 75.55 | 57.38 |
| 9 | 3 | 3 | 100 | 0 | 0 | 77.71 | 57.20 |

*3.6. Comparison with the Spatial-Based Method*

Based on the same sea–land segmentation results, the performance differences between the proposed method and the spatial-based method (the comparison method) were compared. Figure 8 shows the experimental results of Data 2, 3, and 7 in Table 1, where Figure 8a1–d1 show the results of the Fuzhou data, Figure 8a2–d2 the results of the Zhanjiang data, Figure 8a3–d3 the results of the TerraSAR-X Singapore data, Figure 8a1–a3 the results of sea–land segmentation, Figure 8b1–b3 the detection results of the spatial method, and Figure 8c1–c3 the detection results of the proposed method, where the different detection regions are marked in blue boxes. Comparing Figure 8b1,c1, it can be found that there was a false alarm target in Data 2 for the comparison method. Figure 8d1 shows the local enlargement of the false alarm target in Figure 8b1. We can find that the false alarm target is an artificial breakwater according to the map. Comparing Figure 8b2,c2, it can be found that there is a false alarm target in Data 3 for the comparison method. Figure 8d2 shows the local area enlargement of the false alarm target in Figure 8b2. We can find that the false alarm target is a farming dam according to the map. Figure 8a3 shows the segmentation result of Data 7. It can be observed that the sea region of these data includes three parts, the upper, middle, and lower parts, and the sea-crossing bridge is distributed in the right bifurcation of the middle water, which increases the difficulty of detection. Comparing Figure 8b3,c3, we can find that the sea-crossing bridge is not correctly detected using the spatial-based method, while the proposed method obtained the correct detection. Figure 8d3 shows the enlarged view of the local area of the bridge in Figure 8c3. Because the bridge is located in the middle water, which was not correctly extracted, a missed detection occurs in the comparison method. Table 5 shows the performance comparison between the proposed method and the spatial method. It can be observed that the proposed method outperformed the comparison method for all three data.

**Table 5.** Comparison of the bridge detection results between the proposed method and the spatial method.

| Data | Num | The Comparison Method | | | | The Proposed Method | | | |
|---|---|---|---|---|---|---|---|---|---|
| | | Correct | Pd (%) | False Alarm | Pf (%) | Correct | Pd (%) | False Alarm | Pf (%) |
| 2 | 1 | 1 | 100 | 1 | 50 | 1 | 100 | 0 | 0 |
| 3 | 2 | 2 | 100 | 1 | 33.3 | 2 | 100 | 0 | 0 |
| 7 | 1 | 0 | 0 | 0 | 0 | 1 | 100 | 0 | 0 |

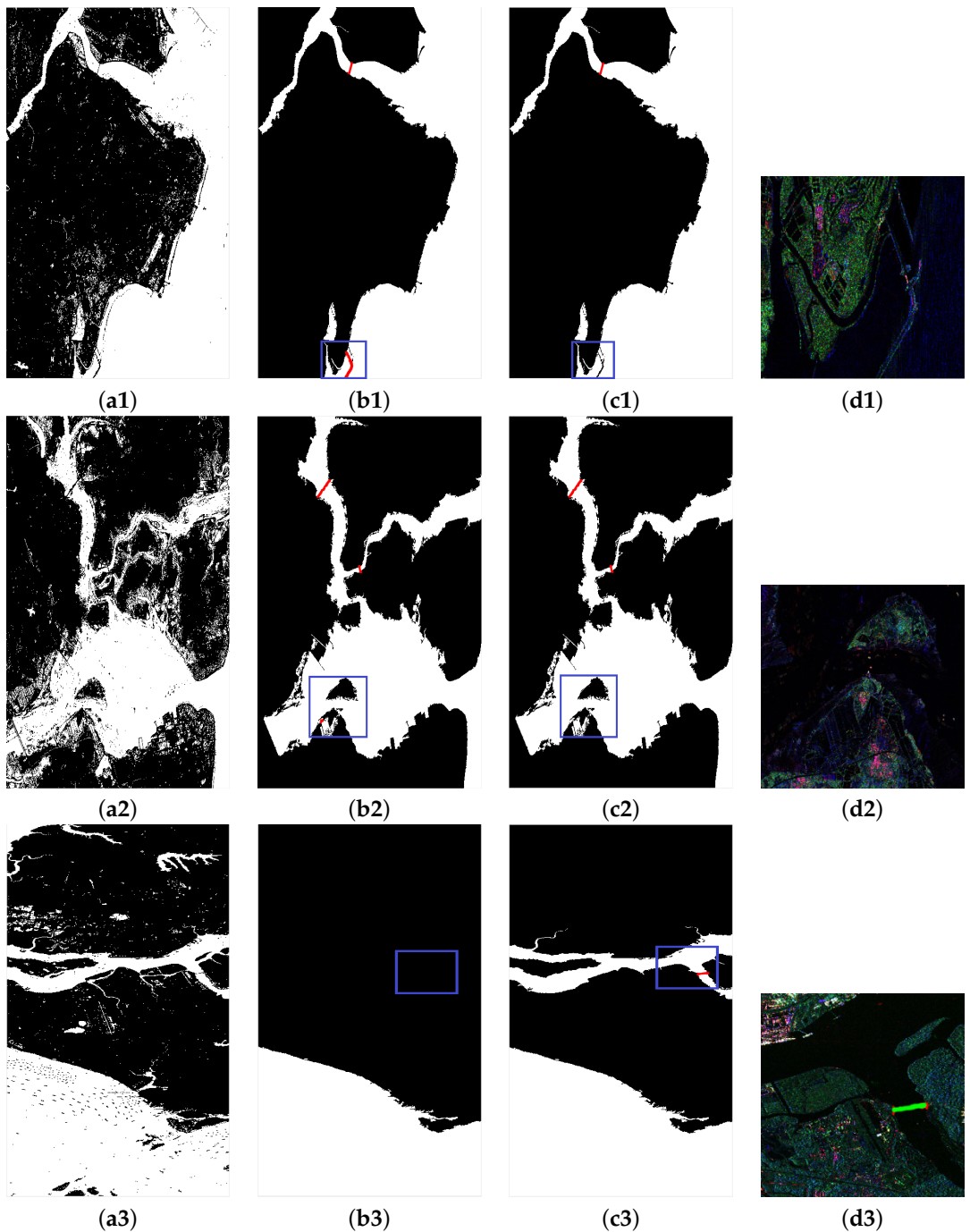

**Figure 8.** Comparison of the bridge detection results between the proposed method and the spatial method. (**a1**–**a3**) Sea–land segmentation results. (**b1**–**b3**) Results of the spatial detection methods. (**c1**–**c3**) Results of the proposed method. (**d1**–**d3**) The local enlargement of the different detection regions marked in blue boxes in (**b1**–**b3**) and (**c1**–**c3**). (**a1**–**d1**) Results of the Fuzhou data. (**a2**–**d2**) Results of the Zhanjiang data. (**a3**–**d3**) Results of the TerraSAR-X Singapore data.

## 4. Discussions

Taking full advantage of the spatial structure and polarization scattering characteristics of the sea-crossing bridge, this paper proposed a sea-crossing bridge detection method based on the windowed level set sea–land segmentation and polarization parameters' discrimination. The method innovatively uses the joint distribution of windowed regions to measure the distribution of a single-pixel coherent matrix in the level set segmentation, uses the polarization similarity parameter to measure the similarity of water regions, and

uses the polarization entropy and scattering angle parameters to distinguish the bridge from the false alarm target. The method enables the detection of sea-crossing bridges to be applied to the detection of different bridges with different polarimetric SAR data in different bands and different scenes, and the false alarms of the traditional spatial method were effectively avoided.

When the level set segmentation method based on the statistical distribution of the single-pixel coherent matrix was used for the sea–land segmentation of the SAR images, the varied scattering matrix may lead to many isolated pixels or small regions in the segmentation results since the sea and land regions are not homogeneous regions. This will lead to the segmentation of images into strongly scattering and non-strongly scattering regions or incorrect segmentation results because the coastal low-scattering region is connected with the sea. Using the joint multi-pixel distribution of window regions instead of the single-pixel distribution can effectively avoid the problem and, thus, ensure the correctness of sea–land segmentation. The segmentation results of the TerraSAR-X San Francisco data in Experiment 4 with different window sizes showed that the erroneous segmentation was gradually reduced with the increase of the window sizes.

The method of water merging only based on the contour distance may incorrectly merge some low-scattering regions. The error can be eliminated by a fusion of merging by the polarization similarity parameter and contour distance. The bridge ROI detected by water merging may be natural terrains and coastal farming regions. By a statistical measurement of the polarization entropy and scattering angle parameters, the false alarms can be effectively eliminated. The comparison results between the proposed method and the spatial method in Experiment 5 demonstrated the phenomenon.

Assuming that the size of the image is $n \times n$, when performing the sea–land segmentation, if the number of iterations of level set segmentation is $k$, then the time complexity of the traditional level set algorithm is $O(kn^2)$, while using the windowed level set segmentation algorithm, if the window size is $w \times w$, the algorithm time complexity is $O(kw^2n^2)$. According to Equation (12), considering that the average coherent matrix of each pixel window region can be calculated and stored in advance, the windowed level set segmentation algorithm's time complexity is still $O(kn^2)$. When performing water merging, if the number of water branches is $N_c$, the time complexity of Algorithm 1 is $O(n)$. Because the number of the main water regions is limited, the time complexity of Algorithm 2 is also $O(n)$. When performing the bridge detection, since there is a double loop for Algorithm 3, the time complexity of the ROI extraction is $O(n^2)$. When performing the ROI recognition using the $H/\alpha$ parameter, only the $H/\alpha$ of the pixels in the ROI need to be calculated, so the computation time can be neglected. In summary, the main computation time of the proposed method is focused on the windowed level set segmentation, and the time complexity is $O(kn^2)$.

## 5. Conclusions

From the perspective of improving the performance of sea–land segmentation and fusing the polarization scattering features of bridges, this paper proposed a polarimetric SAR image detection method for sea-crossing bridges based on windowed level set segmentation and polarization parameter discrimination. The single-pixel probability distribution was replaced by the multi-pixel joint probability distribution in a window region to avoid the erroneous segmentation of the two-region level set segmentation based on the single-pixel distribution. Through the water merging by fusing the polarization similarity parameters and contour distances, the sea–land segmentation result related to the sea-crossing bridge detection was refined and the wrong segmentation result was eliminated. Based on the segmented sea regions, the ROIs of the bridges were simply extracted by merging the close water regions by distance. By the distribution of the polarization entropy and scattering angle parameters of the extracted candidate ROIs, the false alarm targets formed by natural terrains and farming regions were eliminated. The experimental results of multiple polarimetric SAR data from the San Francisco, Singapore, and Fuzhou

and Zhanjiang coastal regions in China demonstrated the effectiveness of the proposed method. The proposed method can achieve 100% correct detection of sea-crossing bridges with different scenes and different morphologies in different bands with a false alarm rate of 0, and the bridge recognition IoG index was higher than 85%, while the IoU index was around 70%.

**Author Contributions:** Conceptualization, C.L. (Chun Liu) and C.L. (Chao Li); methodology, C.L. (Chun Liu); software, C.L. (Chun Liu) and C.L. (Chao Li); validation, C.L. (Chun Liu) and C.L. (Chao Li); formal analysis, J.Y.; investigation, L.H.; resources, J.Y.; data curation, L.H.; writing—original draft preparation, C.L. (Chun Liu); writing—review and editing, J.Y.; visualization, C.L. (Chun Liu); supervision, J.Y.; project administration, C.L. (Chun Liu); funding acquisition, C.L. (Chun Liu) and J.Y. All authors have read and agreed to the published version of the manuscript.

**Funding:** This work was supported in part by the National Natural Science Foundation of China (NSFC) (Nos. 62101456 and 62171023), in part by the Open Fund of Science and Technology on Electromagnetic Scattering Key Laboratory under Grant 622202Y040104, in part by the Doctoral Mass Entrepreneurship and Innovation in Jiangsu Province (JSSCBS20220936), and in part by the 2022 Suzhou innovation and entrepreneurship leading talents program (Young innovative leading talents) under Grant ZXL2022459.

**Data Availability Statement:** Please contact Chun Liu (liuchun@nwpu.edu.cn) for access to the data.

**Acknowledgments:** The authors would like to thank the Reviewers for their valuable comments and suggestions. The data were provided by the laboratory of polarimetric radar and remote sensing applications of Tsinghua University and the China National Satellite Ocean Application Service.

**Conflicts of Interest:** The authors declare that there is no conflict of interest regarding the publication of this paper.

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
