# Peer review of "Sea-Crossing Bridge Detection in Polarimetric SAR Images Based on Windowed Level Set Segmentation and Polarization Parameter Discrimination"

_remotesensing, doi:10.3390/rs14225856_

Round 1

Reviewer 1 Report

Dear Authors,

Your work is very interesting and the search for new methods of object detection and spatial data set analysis is very much welcome. However, I do have some things that should be taken into account before the final results of your work will be finally published. Below I present them in points.

1. Why are so outdated images used? In addition, there are newer systems than those described in this article, such as Sentinel-1. Please refer to this issue in the article.

2. Please explain what the terms"Nf" and "Nd" mean in table 5.

3. The reference to figure 8 is in Chapter 3.6, figure 8 itself is in Chapter 4. This is confusing to read. Need to correct it.

4. In chapter "3.6. Comparison with the Spatial-based Method" a comparison of the developed method and existing methods was made. It is the right thing to do and one of the main goals of all the work. However, the authors used three different scenes (cases). In order to properly compare the tested methods, it would be necessary to perform the bridge detection on the same image, changing only the detection method. The comparison made in this way does not make it possible to say unequivocally which method is better. This comparison should be corrected.

Reviewer 2 Report

See attached PDF.

Reviewer 3 Report

In this paper, the author proposes a polarimetric SAR image detection method for sea-crossing bridges based on windowed level set segmentation and polarization parameter discrimination. There are two main innovations. First, the method replaces the single-pixel statistical distribution in the traditional level-set segmentation method with a joint statistical distribution of pixel-centered window regions. Second, the bridge regions of interest are discriminated by the polarimetric parameters of the polarization entropy and scattering angle. Generally, it seems to be an efficient method for detecting sea-crossing bridges.

Detailed comments are:

1. In Section 2.1, is equation (4) derived from equation (2) and equation (3), and if so, whether equation (3) should be an expression of likelihood function.

2. In Section 3.5, only the detection performance of window 1 and window 3 is compared with the detection performance of window 5, so what about the detection performance of window 7 or window 9? The current results don't seem to prove that window 5 is the best.

3. The detection rate of the method proposed by the authors can reach 100%. is it because there are fewer test pictures? If there are more SAR images, can the detection rate still reach 100%?

4. Some recent literatures in the past 3 years are missing for fair comparison.

Round 2

Reviewer 2 Report

See attached PDF.

Author Response

Dear Reviewer 2:

Thank you for your valuable comments on our manuscript. After carefully studying your comments, we have revised the manuscript. Some clarifications for your comments are listed as follows.

[Reviewer 2 Comment 1]:

The reviewer has checked the revised above manuscript together with the authors’ reply. The reviewer has accepted the authors’ comments except that regarding with the introduction of back-side (left-right reversed) images (Response to Comment 1).

The reviewer understands that the present method is valid for such back-side images. However, the authors have to add note in the manuscript describing that these images are back-side due to the acquisition method of the data. Otherwise, such images lead confusion to many readers.

[Response to Comment 1]:

Thanks very much for your professional comments. The clarification for the back-side images was added in the Section 3. “Because the acquisition track of the testing TerraSAR-X data is different from AirSAR and RADARSAT-2 data, the figures of the data 6 and data 7 in Figure 3 are back-side compared with the corresponding Google map”.

Yours sincerely,

Chun Liu

Nov 12, 2022

Reviewer 3 Report

I find that the author spends considerable effort to deal with the issues raised.  As a result the paper has been greatly improved and I have no hesitation in recommending it for publication.

Author Response

Dear Reviewer 3:

Thank you for your valuable comments on our manuscript. After carefully studying your comments, we have revised the manuscript. Some clarifications for your comments are listed as follows.

[Reviewer 3 Comment 1]:

I find that the author spends considerable effort to deal with the issues raised. As a result the paper has been greatly improved and I have no hesitation in recommending it for publication.

[Response to Comment 1]:

Thanks very much for your professional comments. An extra reference was added in the introduction part. Some errors in the reference were revised.

Yours sincerely,

Chun Liu

Nov 12, 2022